# Deep mutational scanning of influenza A virus neuraminidase facilitates the identification of drug resistance mutations *in vivo*

Sihan Wang,[1,2] Tian-hao Zhang,[3] Menglong Hu,[1] Kejun Tang,[4] Li Sheng,[5,6] Mengying Hong,[1] Dongdong Chen,[7] Liubo Chen,[8] Yuan Shi,[5] Jun Feng,[5] Jing Qian,[9] Lifeng Sun,[10] Kefeng Ding,[10] Ren Sun,[1,5,11] Yushen Du[1]

**ABSTRACT**  Neuraminidase (NA) is a pivotal surface enzyme and a key therapeutic target in combating the influenza A virus. Its evolution can lead to potential zoonotic transmission, seasonal epidemics, and the emergence of drug-resistant mutants. To gain comprehensive insights into the mutational effects and drug resistance profiles of NA, we employed a high-throughput profiling system to quantify the replication capacity of NA mutants at the single-nucleotide level in mouse lung tissues. The fitness of NA mutants is generally correlated with natural mutation occurrence and is constrained by both the requirement to maintain protein stability and NA function. Leveraging this system, we profiled the drug resistance to the three most commonly used neuraminidase inhibitors (NAIs): zanamivir, oseltamivir, and peramivir. In addition to identifying previously reported drug resistance mutations, we validated novel mutants. Notably, we identified an allosteric mutation that confers resistance to all three drugs, which may affect drug binding by interfering with the tetramerization of NA. Moreover, the fitness cost associated with drug-resistant mutations may limit their widespread dissemination. In summary, we provided a parallel characterization of NA's fitness and drug resistance landscape in an *in vivo* context, which may guide the rational selection of antiviral drugs for optimal therapeutic efficacy and second-generation NAI development.

**IMPORTANCE**  NA is a crucial surface antigen and drug target of influenza A virus. A comprehensive understanding of NA's mutational effect and drug resistance profiles *in vivo* is essential for comprehending the evolutionary constraints and making informed choices regarding drug selection to combat resistance in clinical settings. In the current study, we established an efficient deep mutational screening system in mouse lung tissues and systematically evaluated the fitness effect and drug resistance to three neuraminidase inhibitors of NA single-nucleotide mutations. The fitness of NA mutants is generally correlated with a natural mutation in the database. The fitness of NA mutants is influenced by biophysical factors such as protein stability, complex formation, and the immune response triggered by viral infection. In addition to confirming previously reported drug-resistant mutations, novel mutations were identified. Interestingly, we identified an allosteric drug-resistance mutation that is not located within the drug-binding pocket but potentially affects drug binding by interfering with NA tetramerization. The dual assessments performed in this study provide a more accurate assessment of the evolutionary potential of drug-resistant mutations and offer guidance for the rational selection of antiviral drugs.

**KEYWORDS**  deep mutational scanning, drug-resistant mutations, influenza, neuraminidase

Influenza viruses represent an ongoing public health threat, causing acute respiratory disease with high morbidity and mortality (1). The primary drivers of influenza

Address correspondence to Ren Sun, sunren@westlake.edu.cn, or Yushen Du, lilyduyushen@zju.edu.cn.

Sihan Wang, Tian-hao Zhang, Menglong Hu, and Kejun Tang contributed equally to this article. The co-first authorship order was determined by group discussion.

The authors declare no conflict of interest.

evolution are continuous genetic changes in the two glycoproteins, hemagglutinin (HA) and neuraminidase (NA). HA binds the sialic acids on the surface of the target cell and primes virus entry (2). NA cleaves the newly formed budding virions, enabling viral release (3, 4). Understanding the mutational effects of the genetic changes on HA and NA would enable us to evaluate the viral replication capacity and predict the possible impact of mutations on seasonal epidemics.

We and others have integrated the mutagenesis system with high-throughput sequencing to explore the mutational effects of HA in cell culture. These investigations have characterized antibody escape mutations, reconstructed the evolutionary history of antigenic epitopes, and identified potential vaccine targets on HA (2, 5, 6). However, the mutational effect on NA has received comparatively less attention. In addition to its well-studied catalytic activity, NA was reported to facilitate virus spread through the airway epithelium mucus by cleaving (2-6)- or α (2-3)-ketosidic linkage (7–11). It also interacts with innate immune pathways by mediating the recognition of the complement system (12). Recent studies have identified NA as a potential antigen for elucidating broad neutralizing antibodies (13, 14). Some of these functions of NA would be hard to accurately measure in a cellular model, underscoring the importance of investigating the mutational effects of NA in an *in vivo* context. However, in contrast to *in vitro* screening, which can accurately control factors such as the number of cells and multiplicity of infection (MOI), *in vivo* screening encounters a stronger infection bottleneck. Therefore, optimizing the complexity of the library and the infection scheme becomes crucial to ensure reliable profiling results and maintain library complexity *in vivo*.

To date, NA is the primary drug target for influenza virus treatment. Neuraminidase inhibitors (NAIs), such as zanamivir (ZA), oseltamivir (OS), and peramivir (PE), have become widely used antiviral agents for the management and prevention of influenza infections (15–18). The three NAIs share similarities in their binding mechanisms, which contain a carboxylate group that interacts with the trinity composed of the arginyl side chains of NA R118, R292, and R371 and an N-acetyl group that interacts with the hydrophobic pocket of NA comprised principally by the side chains of I222 and W178 (19). Nevertheless, their unique characteristics lead to different drug-resistance profiles (20). With the extensive clinical applications of NAIs, the emergence of drug-resistance mutations has become a concern. Various approaches have reported and validated several drug-resistant mutations, such as isolating and sequencing drug-resistant influenza A virus (IAV) variants from the field or continuously passaging wild-type (WT) viruses under drug selection pressure (18, 21–30). Most of these drug-resistant mutations entail a single nucleotide alteration that reduces the binding capacity of NA to NAIs, resulting in reduced therapeutic efficacy (20, 21). Therefore, a comprehensive characterization of the drug resistance profiles of different NAIs is critical for the rational selection of appropriate antiviral drugs to combat resistance in the clinic and facilitates the optimization of next-generation NAIs.

In this study, we established a deep mutational profiling system for NA. By introducing a library of single-nucleotide mutants of NA into mouse lung tissues and employing deep sequencing, we quantified the replication fitness of each mutant. To ensure the reliability of our fitness profiling data set, we controlled the complexity of the mutant library, the infection dose, and the timing of sample collection from infected mouse lungs. Compared with screening in cell culture, fitness profiling in the *in vivo* context represents a more biologically relevant condition and provides information on immune-sensitive mutations.

Using this system, we characterized the resistance profiles of OS, ZA, and PE *in vivo*. We identified and validated several new drug-resistant mutants by comparing the mutational effects on viral replication under both drug selection and no drug selection conditions. Interestingly, we found an allosteric drug-resistant mutation (V205I) outside the drug-binding pocket on the NA tetramerization interface. Finally, we analyzed the global dynamics of NA variants and noted that the frequency of NAI-resistant mutations

did not increase significantly over the last decade, probably due to the impact of these mutations on viral replication fitness.

## RESULTS

### Systematic profiling of the mutational effect of the neuraminidase of influenza A/WSN/33 (H1N1) viruses by *in vivo* screening

To examine the influence of NA mutations on viral fitness *in vivo*, we established a deep mutational screening system in a mouse model (Fig. 1A). We ensured sufficient replication of all variants in mouse lung tissues by controlling the complexity of the viral mutant library. To do so, we divided the NA segment into five sub-libraries, each containing single nucleotide mutations in the 240 bp range, which covered amino acids 7–407 (Fig. S1A). Random mutagenesis introduced the mutations into each 240-bp region, where the number of templates was controlled to maximize the percentage of single nucleotide mutations (Fig. S1B). In this way, the library complexity is estimated to be ~$10^3$ magnitudes, ensuring the relatively high frequency of each mutation and minimizing the stochastic effect of the emergence of new mutations. The 240-bp region can be covered by both forward and reverse reads of Illumina NovaSeq PE250, allowing pair-end sequencing error correction to distinguish sequencing errors from true mutations in the library.

The reconstitution of virus libraries was performed in 293T cells by co-transfecting each mutant NA plasmid library with seven WT plasmids encoding other influenza segments (22, 23). Subsequently, the virus libraries underwent amplification in Madin-Darby Canine Kidney (MDCK) cells before being utilized for *in vivo* infection with an MOI of 0.1 for 48 h. It is worth noting that the amplification step in MDCK cells may introduce selective pressure on viral mutants; however, it produced sufficient viral titers for subsequent *in vivo* experiments and provided a strict linkage between viral genotype and surface phenotype (24). We used $1 \times 10^5$ $TCID_{50}$ viruses for intranasal infection, ensuring sufficient coverage of each mutant. The viral growth curve of WT viruses ($1 \times 10^5$ $TCID_{50}$) in mice peaked on days 2 to 4 after infection, so we harvested lung tissues on day 3 to select the virus libraries (Fig. S1C). Biological triplicates consisting of three mice were included for each NA sub-library. The mutant DNA plasmid and passaged virus libraries harvested from lung tissues were simultaneously subjected to next-generation sequencing (NGS) analysis. Mutations with a frequency lower than 0.01% in the DNA library were filtered out, including 2,646 single-nucleotide mutations, covering approximately 88.5% of the total NA residues (401/453). Each mutation's relative fitness (RF) score was calculated as the proportion of the relative frequency in the passaged library to that in the DNA library (Table S1). A strong correlation of RF scores among different mice (biological replicates) was obtained, indicating the reproducibility of our approach (Fig. 1B). Relative fitness scores of synonymous mutations were centered around 1 ($\log_{10}$ RF = 0), while a clear separation in the distribution of RF scores was observed between synonymous mutations and nonsense mutations (Fig. 1C). Furthermore, we observed specific tolerance to mutations in different domains of the NA protein. The transmembrane and stalk domains, which were known to exhibit higher variation across different NA subtypes and susceptibility to insertions or deletions, displayed greater tolerance to mutations (Fig. S1D) (25–27). These results demonstrated the validity of our method for generating a comprehensive annotation of the impact of each nucleotide substitution on NA protein in an *in vivo* context.

To examine if the fitness of NA mutants correlated with their natural occurrence, we analyzed the sequence diversity of 12,667 human N1 NA sequences (2010–2020) from the Influenza Research Database (IRD). The entropy at each site was generally correlated with the RF score measured in our screening (Fig. 1D and E, Spearman's $\rho = 0.66$, $P < 0.0001$). Amino acid positions that incurred a high fitness cost upon mutation tended to be highly conserved. The correlation between natural occurrence frequency and RF score was also significant at each mutation level (Fig. 1F, Spearman's $\rho = 0.59$, $P < 0.0001$).

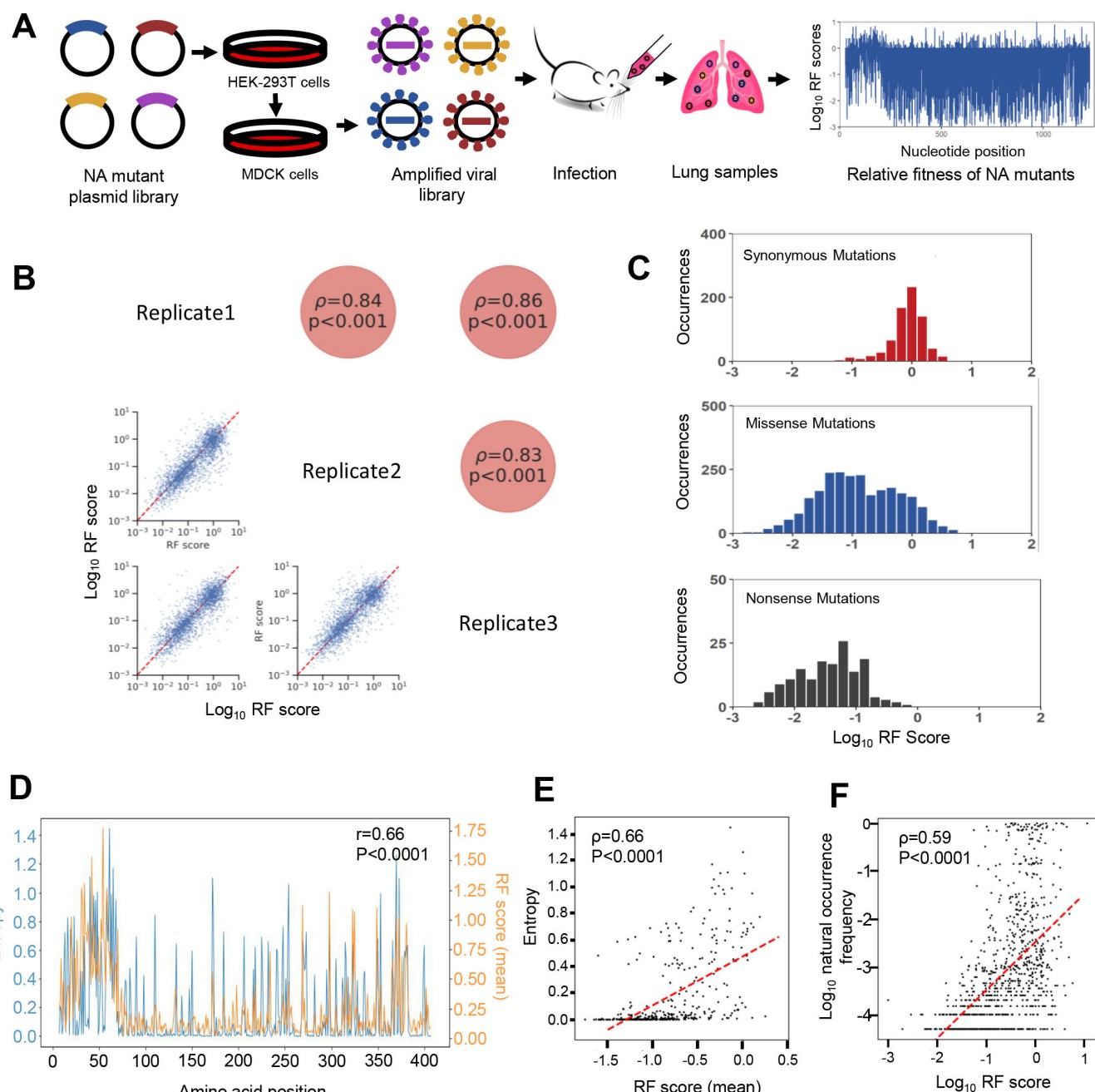

FIG 1  Fitness profiling of NA single nucleotide mutations by *in vivo* screening. (A) Schematic plot illustrating the experimental procedure. Each mutant viral library was reconstituted by co-transfecting the mutant NA plasmid library with seven plasmids encoding the remaining wild-type viral fragments in HEK-293T cells. The mutant viral libraries were then amplified in MDCK cells. Mice were intranasally infected with the mutant viral libraries, and lung samples were collected at day 3 post-infection. Viral RNA and the mutant NA plasmid library were extracted and subjected to NGS. The RF scores of each mutant in the library were quantified. (B) Correlation of RF scores of NA mutations from biological triplicates (Spearman correlation test). (C) Distribution of the RF scores for synonymous, missense, and nonsense mutations. (D) The correlation between entropy and RF scores at each NA residue is depicted. The orange line represents the mean RF score of all mutations at each residue, while the blue line represents the Shannon entropy. The Shannon entropy was calculated using 12,667 human N1 NA sequences from 2010 to 2020 obtained from the IRD database. (E) The correlation between the mean RF score and entropy for each amino acid position of NA (Spearman correlation test). (F) The correlation between the RF score and the frequency of natural occurrence for individual NA mutants (Spearman correlation test).

However, there were instances where discrepancies emerged between entropy and RF scores.

On the one hand, some residues near the NA catalytic pocket (242, 257, 258, and 288, N1 numbering) showed relatively high entropy but were deleterious upon mutation. This discrepancy could be attributed to compensatory mutations in other influenza strains or selection pressures beyond NA activity alone, such as the interplay between HA and NA activity (28–30). On the other hand, certain residues (for example, 27, 148, and 420, N1 numbering) showed high mutation tolerance based on our screening but were conserved in the database, suggesting that the sampling of natural strains in the database may have been insufficient to represent the full evolutionary landscape completely. Thus, *in vivo* NA fitness profiling may provide a more comprehensive evaluation of the evolutionary potential of NA mutants.

## NA fitness profiling reveals functional sites essential for complex formation or immune evasion

To explore the biophysical basis of mutational effects, we performed protein stability simulations using PyRosetta (31, 32). At the residue level, we found a general correlation between the impact on NA monomer stability (ΔΔG) and the RF scores (Fig. 2A). Consistent with the threshold model (33, 34), all of the observed mutations at residues that had a major impact on protein stability (median ΔΔG >10) were highly deleterious (median RF <0.2). Interestingly, certain residues with minimal impact on monomer stability exhibited high deleteriousness (Fig. 2A), suggesting functional importance. Notably, these residues included well-known catalytic sites (D151, R152, R225, E277, R292, R368, and Y402; Fig. 2B, N1 numbering), and others were located at the intramolecular interface of the NA tetramer, implying their role in maintaining tetramer integrity and preserving NA protein function (Fig. 2C).

In addition to the biophysical constraints, the fitness effects observed through *in vivo* screening may also reflect the ability of mutants to effectively spread within the lung environment under the selective pressure exerted by the host immune system. To directly demonstrate this, we conducted an additional selection of the same amplified viral library stock in A549 cell culture (*in vitro*, MOI = 0.01, 48 h, Fig. S2A) and compared the results with the *in vivo* findings. The fitness profiling *in vitro* showed a robust correlation among biological replicates and a good separation between synonymous and nonsense mutations (Fig. S2B and S2C). The distribution of silent mutations was highly consistent between *in vitro* and *in vivo* (Fig. 2D, upper panel). However, we observed certain missense mutations that exhibited deleterious effects *in vivo* but were considered neutral *in vitro* (RF *in vitro* >0.5, RF *in vivo* <0.2, fold change >5, *P* < 0.05, Fig. 2D, lower panel). To further validate these mutations, four representative mutants were constructed individually, all of which showed comparable replication capacity with WT viruses in cell culture (Fig. 2E). We assessed the relative replication capacity of the mutants by mixing each mutant with the WT virus in equal proportions to better recapitulate the screening condition. We then conducted intranasal infections in mice and infections in A549 cells. Viral RNA from the input viral mix, A549 cell supernatant, and mouse lung tissues was analyzed using Sanger sequencing. The relative peak heights of the mutant and WT viruses at specific positions were quantified to evaluate their relative replication capacities (Fig. 2F; Fig. S2D). These mutants showed reduced replication capacity compared with WT *in vivo*, consistent with the screening result. Given that the inability to evade the host immune response can contribute to the differences in *in vivo* and *in vitro* fitness of mutants, and considering that the interferon response plays a crucial role in the innate immune response against viral infections, we investigated the interferon sensitivity of the four selected mutations. A549 cells were infected with individual mutants or WT viruses at MOI 0.01 with and without exogenous Type I Interferon (IFN) treatment (IFNa2, 1,000 U/mL). The viral copy number was examined at 24 h post-infection, and the IFN sensitivity was calculated as the fold decrease in viral copy number upon IFN treatment. Notably, one of these mutations, V67G, was significantly more sensitive to IFN selection than WT viruses (Fig. 2G).

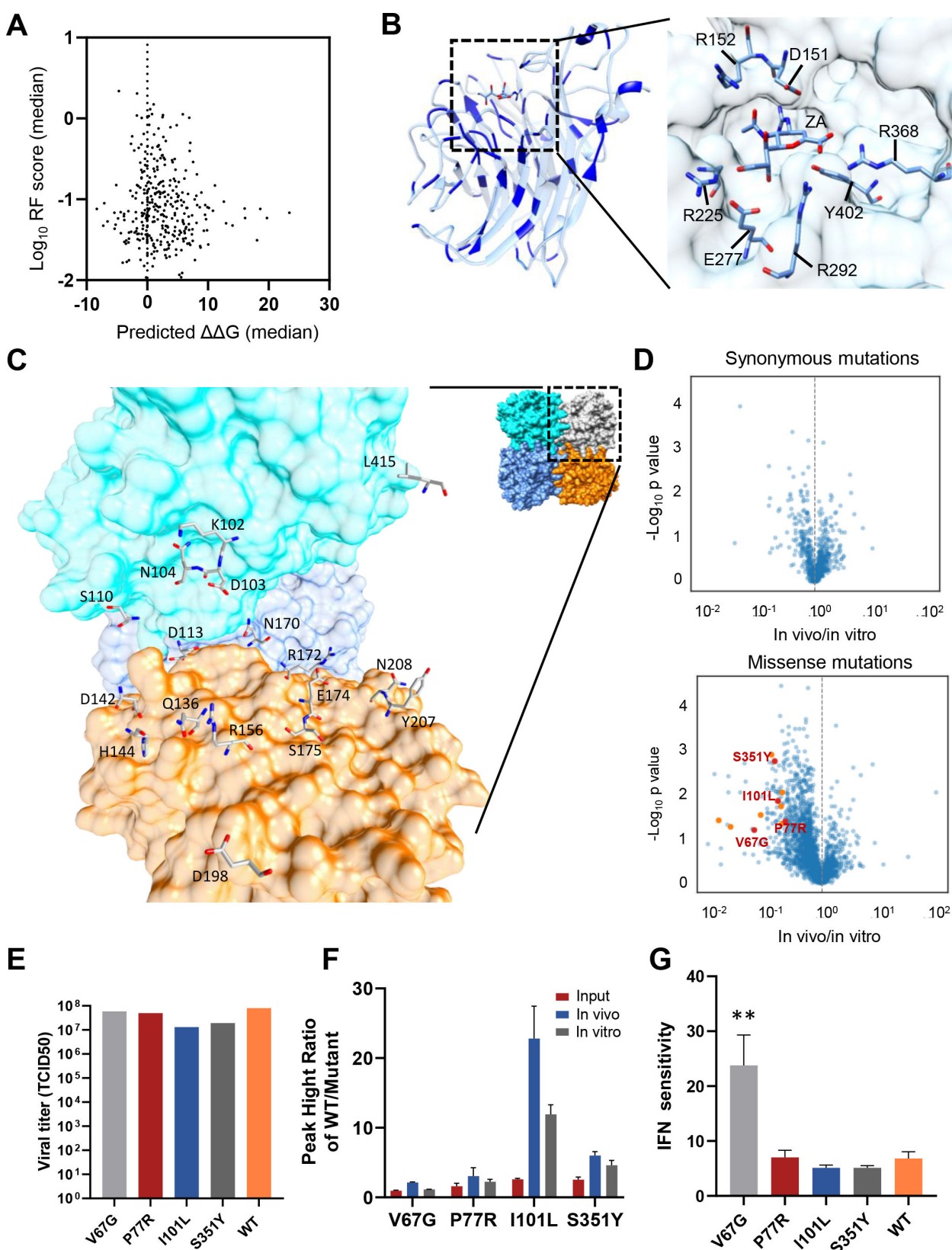

**FIG 2** Fitness profiling of NA reveals functional sites essential for complex formation or immune evasion. (A) The figure illustrates the median predicted ΔΔG (folding free energy change) and the median RF score (relative fitness score) of mutations at each amino acid site in the NA protein. The changes in folding free energy ΔΔG were predicted using PyRosetta based on the NA monomer structure (PDB: 3B7E). The median values were used here instead of the mean values (Continued on next page)

**FIG 2** (Continued)

to provide a more robust assessment by reducing the influence of outliers. (B) Ribbons represent the NA monomer. Functional residues without major impact on protein stability are labeled in blue, with the ones located in the catalytic site region shown with an enlarged view in the right panel, including D151, R152, R225, E277, R293, R368, and Y402 (PDB: 3B7E, N1 numbering). (C) Molecular surface representation of the NA tetramer, with each monomer depicted in distinct colors to highlight the intermolecular interfaces. Functional residues located at inter-molecular interfaces are labeled. (D) The volcano plot of the RF scores of synonymous and missense mutations. The X-axis represents the $\log_{10}$ RF score of each mutation *in vivo* divided by *in vitro*. The Y-axis represents the $-\log_{10}$ *P*-value of each mutation *in vivo* compared with that *in vitro*. Yellow dots represent the mutations with RF *in vitro* >0.5, RF *in vivo* <0.2, fold change >5, and *P* < 0.05. Red dots are the ones picked for further validation. (E) The viral titer of four representative mutants and WT viruses is quantified using $TCID_{50}$. (F) Relative growth capacity of mutant WT virus. The Y-axis represents the relative peak heights of mutant and WT at certain positions, quantified by Sanger sequencing. (G) The interferon sensitivity of four mutations and the WT virus is shown upon exogenous IFN-a2 treatment in A549 cells. The Y-axis represents the fold decrease in viral copy number upon IFN treatment compared to no treatment (*N* = 3). *$P$ < 0.05, **$P$ < 0.01, ***$P$ < 0.001 (two-tailed *t*-test).

The above data suggested that the fitness profiling in the *in vivo* setting provided a comprehensive evaluation of the functional sites of NA, revealing not only the biophysical or catalytic functions of NA but also the potential immune evasion functions.

## Systematically profiling drug-resistant mutations of NA to three NAIs

With the established deep mutational scanning platform in the *in vivo* context, we systematically evaluated the drug resistance of NAIs that were used most widely in the clinic, including OS, ZA, and PE (Fig. 3A). The IC90 dose was selected as the screening concentration for OS and PE (1 mg/kg/d) and ZA (0.1 mg/kg/d). This dose demonstrated an ~10-fold inhibition of WT virus replication while exhibiting limited inhibitory effects on the most frequently observed NAI-resistant mutation, H275Y (18, 35, 36) (Fig. S3A and S3B). The NA virus libraries were also inhibited ~10-fold upon each drug selection. Furthermore, each drug selection inhibited the NA virus library by approximately 10-fold (Fig. S3C). No significant difference in viral inhibition was observed when administering drugs at 2 or 6 h post-infection. Therefore, we selected the 6-h post-infection time point for drug administration to ensure the successful infection of viral mutant libraries (Fig. S3D). Like the RF profiling, we included three biological replicates for each selection condition. For each particular drug, a mutation's drug resistance score (W) was calculated as the proportion of the RF score under drug selection to that without drug selection. A mutation was classified as resistant if it exhibited significantly higher fitness (W) than synonymous mutations under the same experimental conditions (two-tailed *t*-test). Alternatively, if multiple mutations in a single residue displayed a resistance phenotype (more than two mutations within one residue had min W > 1.5 for all three replicates), they were also considered resistant. A total of 99 mutations were identified as resistant to OS (Fig. 3B), including previously reported drug-resistant mutations H275Y and D199G (N1 numbering) (37–39). Conversely, 82 and 86 mutations were resistant to ZA and PE, respectively (Fig. 3B, Table S2). Consistent with previous studies, H275Y showed resistance to OS and PE but not to ZA (40, 41). This result was confirmed with the NA activity assay (Fig. S3E). While H275Y was highly resistant to OS in our data set, other amino acid changes, such as H275N and H275Q, remained sensitive, confirming the reliability of our drug-resistance profiling (Fig. S3F) (41).

Comparing the drug-resistant mutations across the three NAIs, we noticed some positions exhibiting resistance to multiple drugs (Fig. 3C). As the three drugs share similarities in their binding mode with the NA catalytic pocket, mutations conferring potential resistance across 2–3 drugs warrant particular attention. The residues that directly interact with chemicals (for example, R118, D151, and R152) are usually vital for viral replication. Thus, residues that carry indirect interactions may have greater clinical significance. For example, F371V showed resistance to all three drugs. F371 residue directly contributes to the hydrophobic core while stabilizing nearby R368 residue, whose side chain can form strong hydrogen bonds with ligands. Mutations at this site may enhance the flexibility of the loop containing the R368 residue and consequently decrease the binding affinities of ligands (Fig. 3D). H275Y, the most common OS- and PE-resistant mutation, also affects drug binding indirectly by changing the position

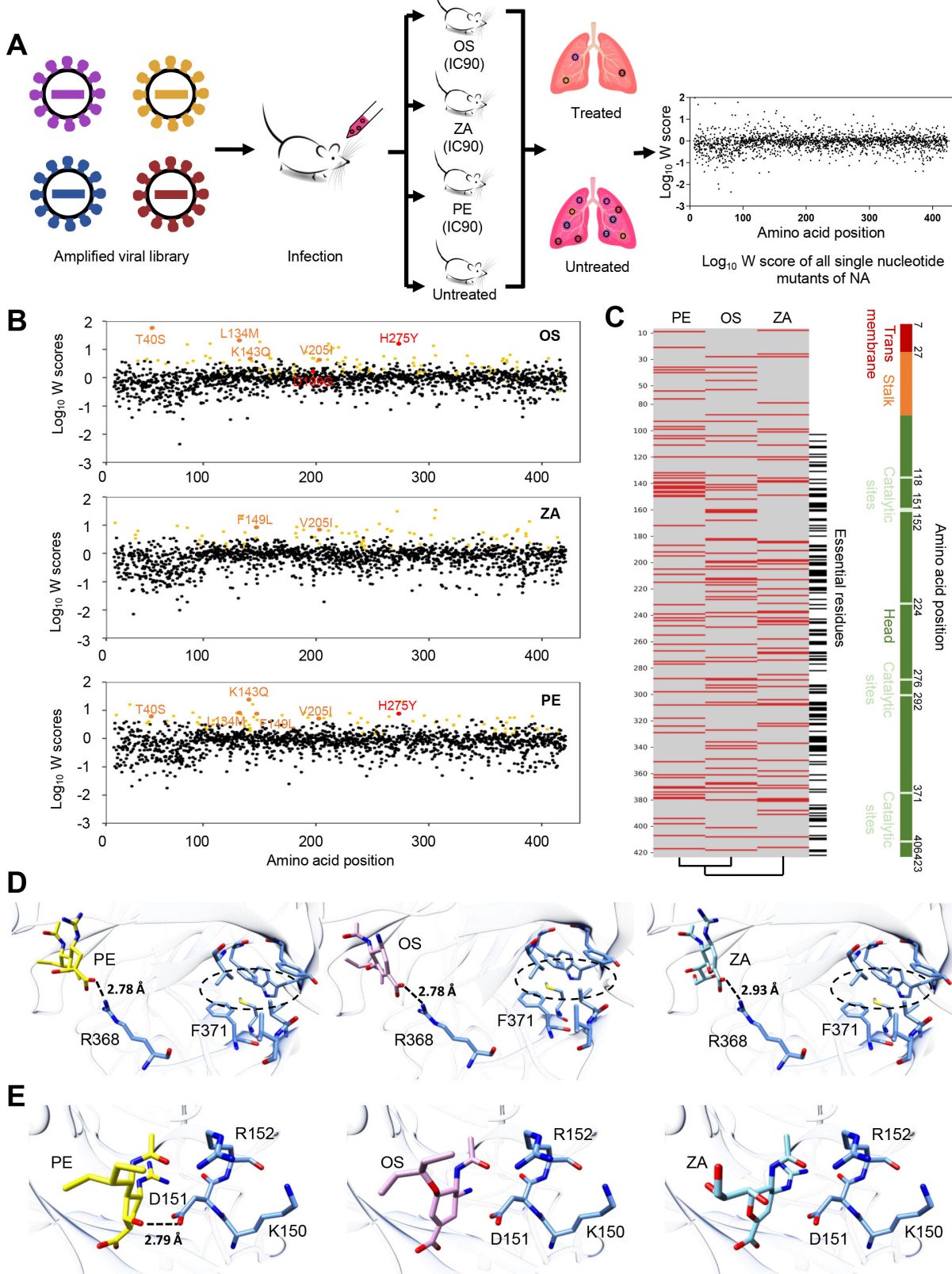

**FIG 3** Systematic profiling of drug-resistant mutations of NA to three NAIs. (A) Schematic plot of the experiment procedure for drug-resistant mutation screening *in vivo*. All three drugs were administered at the concentration of IC90 for WT viruses. OS was given orally, ZA was intranasally, and PE was intramuscular. W scores are calculated as the ratio of the RF score with drug selection to that without. (B) All identified drug-resistant mutations are labeled in (Continued on next page)

**FIG 3** (Continued)

yellow (99 for OS, 82 for ZA, and 86 for PE). The previously reported drug-resistant mutations, D199G and H275Y, are highlighted in red. The newly validated drug-resistant mutations in our screen are highlighted in orange in T40S, L134M, K143Q, F149L, and V205I. (C) Heat map showing the amino acid positions with drug-resistant mutations on NA (indicated by red bars). Lethal residues are depicted with black bars on the right. The domains and catalytic sites are also indicated. (D) Substitutions on F371 spatially affect the drug-binding capacity of R368. (E) The drug-binding capacity of D151 is spatially influenced by substitutions on residue K150, resulting in drug resistance against PE but not OS and ZA.

of E277 (37, 42, 43). In addition to the presence of multidrug-resistant mutations, we identified specific mutations that confer resistance to individual drugs. For example, K150T (N1 numbering) was found to be resistant to PE, which might lead to the disruption of the strong hydrogen bond between D151 and PE but not OS or ZA (Fig. 3E). The identification of these drug-specific mutations may facilitate the rational selection of antiviral drugs in the clinic.

## Validation of representative newly identified NAI-resistant mutants

Besides the literature-reported drug-resistant mutations, we aim to validate new mutations that could potentially confer reduced sensitivity to NAIs. We picked 11 nonreported mutations that confer resistance to 2, or all tested NAIs, and constructed them individually. The six mutations with RF scores less than 0.1 in the control condition were too deleterious to obtain sufficient virus for downstream drug-resistance assays (Fig. S4A). The reconstituted titer of the other five mutations was generally correlated with the RF score, again showing the reliability of our fitness profiling data (Fig. S4B). To validate the drug resistance of these five mutations, we first performed the NA activity assay and calculated the IC50 of NAIs for each mutation. H275Y was included as the positive control (Table 1; Fig. 4A). The newly identified mutations demonstrated varying degrees of resistance to the three NAIs, as shown in Table 1. Among them, V205I and L134M exhibited the strongest resistance phenotype, further confirmed through viral growth assays in cell culture and intra-nasal infection of mice (Fig. 4B; Fig. S4C and S4D).

Furthermore, we tried to elucidate the potential mechanism of drug resistance of the two new mutations from a structural perspective. Similar to other well-recognized drug-resistant mutations, L134 is located at the intra-hydrophobic core, consisting of eight functional residues (Arg118, Asp151, Arg152, Arg224, Glu276, Arg292, Arg371, and Tyr406). The side chain of the mutation L134M may directly alter the catalytic pocket and consequently reject the drug binding (Fig. 4C). Interestingly, the mutation V205I exhibits resistance to multiple drugs despite its location away from the catalytic pocket. V205 is situated at the interface between NA monomers, and this mutation has the potential to disrupt the tetramer structure, resulting in the separation of the two NA monomers (Fig. 4D). Besides the NA activity assay and viral growth assay, we also confirmed the loss of binding of V205I to NAIs through biochemical analysis, using PE as an example. Isothermal titration calorimetry (ITC) results of V205I showed a twofold decrease of Kd in PE compared to WT (Fig. 4E). The thermodynamic parameters obtained in the ITC are shown in Fig. 4F.

## The replication fitness cost of NAI-resistant mutations may limit their dominance among circulating strains

Considering the mounting prescription of NAIs for early influenza infections, it is necessary to monitor the circulating NAI-resistant variants at the population level (44–46). To this end, we analyzed the dynamics of all potential drug-resistant mutations identified in our high-throughput screen using human N1 NA sequences from IRD ranging from 2010 to 2020. The occurrence dynamics of all identified drug-resistant mutants are shown in Fig. 5A (99, 82, and 86 mutations for each drug, respectively), with the mean occurrence shown with a red line. The frequency of drug-resistant mutants did not show a noteworthy increase during the past decade despite the increase in drug usage (44–46). We also highlighted the time trajectory of the major drug-resistant

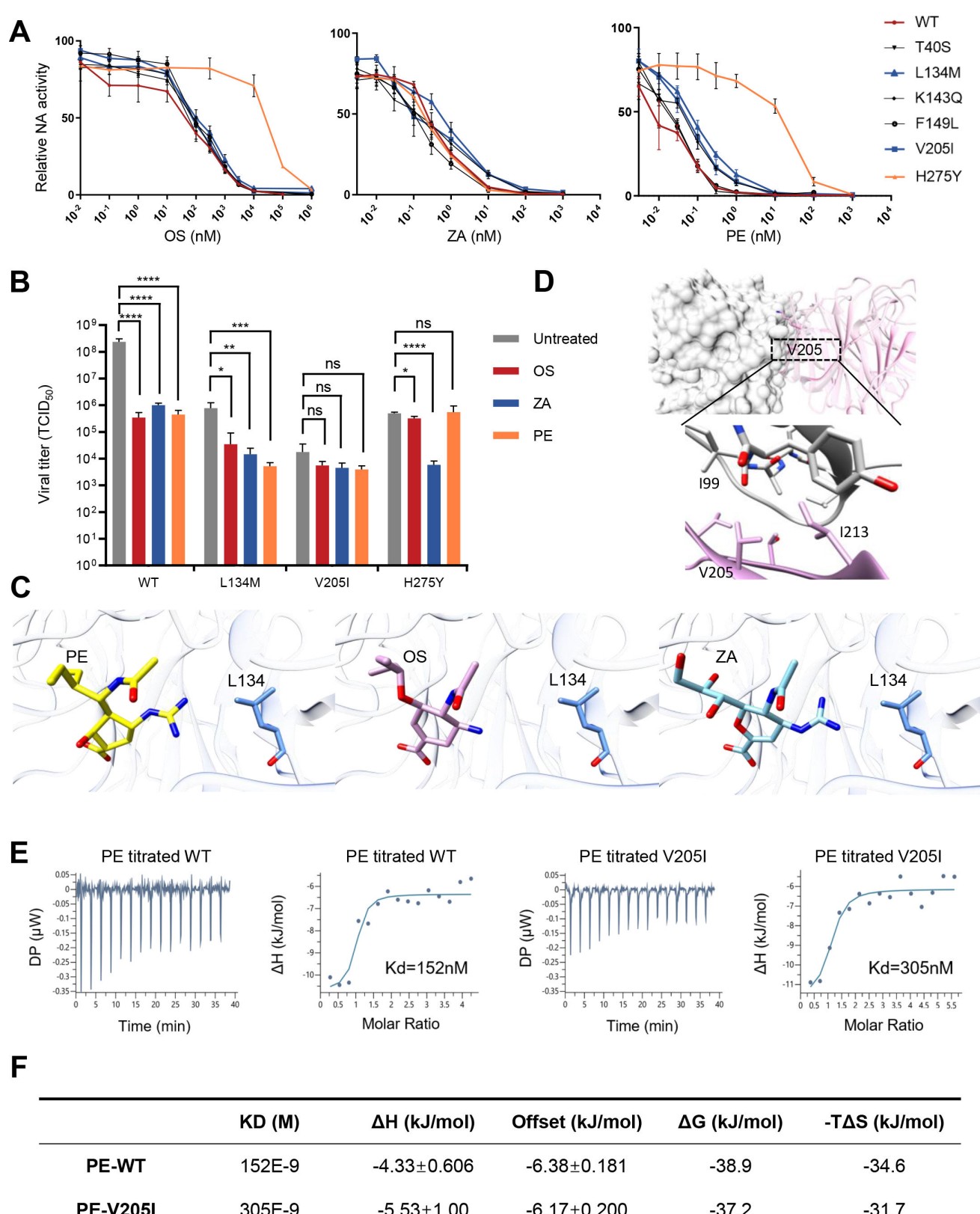

**FIG 4** Validation of representative newly identified NAI-resistant mutants. (A) Validation of drug-resistant mutations by NA activity assay. The relative NA activity is measured as a percentage of the test sample with drug treatment compared to the control sample without. The data represent the average of three independent biological replicates. WT is labeled in red, the reported NAI-resistant mutation (H275Y) is labeled in orange, the newly identified NAI-resistant

**FIG 4** (Continued)

mutations with the strongest phenotype (L134M and V205I) are labeled in blue, and the remaining validated NAI-resistant mutations (T40S, K143Q, and F149L) are labeled in black. Error bars denote SEM. (B) *In vitro* validation of drug-resistant mutations in A549 cells ($N = 3$). The viral titer ($TCID_{50}$) of WT and each mutant under NAI treatment is shown with a bar plot. Error bars denote SD. $*P < 0.05$, $**P < 0.01$, $***P < 0.001$, $****P < 0.0001$, ns: not significant (two-tailed *t*-test compared with the untreated group). (C) NA residue L134 is near the NAI binding sites. The substation of M on L134 may interrupt the drug binding. (D) V205 is on the inter-molecular interface (PDB: 5NZE). The lower panel provides a detailed spatial view of V205. (E–F) PE binding affinity to WT and V205I NA is measured by ITC. Both raw ITC data and the fitted single-binding-site model are shown.

mutations conferring resistance to two or all tested NAIs reported or validated in the present study (Fig. 5B). Consistently, they did not show a trend of escalation.

Given that our platform enables the simultaneous assessment of replication fitness and drug resistance, we investigated whether the absence of drug-resistant mutations was attributable to their impact on viral replication. Indeed, upon plotting the RF scores against the degree of resistance (W), we observed that nearly 97% of mutations conferring resistance to the three NAIs exhibited compromised replication capacity (Fig. 5C), including all the validated ones, as we highlighted. The diminished fitness of drug-resistant mutations, such as L134M, V205I, and the previously reported H275Y, may contribute to the limited dominance of these mutations at the population level. We confirmed the reduced fitness of these newly identified drug-resistant mutants through assessments of NA activity and viral titers in both *in vitro* and *in vivo* settings (Fig. 5D).

## DISCUSSION

In the current study, using a mouse model, we employed a high-throughput genetics platform to map the fitness cost and drug resistance of single-nucleotide mutations of influenza A NA. Our findings revealed a strong correlation between fitness effects and the natural occurrence of mutations, which could be attributed to protein stability and functional constraints. The drug resistance profiling confirmed previously reported mutations and validated new mutations, including one with a potential allosteric regulatory function. Finally, we analyzed the dynamics of recent occurrences of drug-resistant mutations and elucidated their limited global spread by considering their fitness cost in the absence of drug pressure.

To the best of our knowledge, our study is one of the first deep mutational scans performed in an animal model at single nucleotide resolution. Prior to comprehensive profiling of mutations to a target protein, it was performed in cell culture or at the protein level (6, 47–49). However, in an infection process, viruses need to penetrate the airway mucus layer to access epithelial cells, counteract host innate and adaptive immune responses, and transmit to other tissue locations, all of which cannot be recapitulated with these *in vitro* systems (50, 51). Thus, our *in vivo* system provides a more comprehensive evaluation of the replication capacity of mutations. Differing from the *in vitro* system, the *in vivo* infection may face a stronger infection bottleneck with

**TABLE 1** NAI susceptibility of influenza A/WSN/33 (H1N1) virus carrying single amino acid substitutions of NA[a]

| NA mutations | IC50 (mean ± SD) (nM) | | |
|---|---|---|---|
| | OS | ZA | PE |
| WT | 22.45 ± 18.13 | 0.16 ± 0.07 | 0.01 ± 0.00 |
| **T40S** | **48.84 ± 29.48** | **0.07 ± 0.04** | **0.01 ± 0.01** |
| **L134M** | **97.09 ± 48.55** | **0.28 ± 0.12** | **0.05 ± 0.02** |
| **K143Q** | **63.37 ± 46.05** | **0.11 ± 0.05** | **0.03 ± 0.01** |
| **F149L** | **98.65 ± 26.91** | **0.08 ± 0.03** | **0.01 ± 0.00** |
| **V205I** | **86.09 ± 43.70** | **0.20 ± 0.12** | **0.05 ± 0.03** |
| H275Y | 22550 ± 7452 | 0.12 ± 0.07 | 2.00 ± 1.00 |

[a]IC50 represents the NAI concentration that is able to inhibit 50% of the NA activity. Amino acid numbering is based on N1 NA. Values are means ± SD from three independent experiments. The newly identified drug-resistant mutations in our assay are boldfaced.

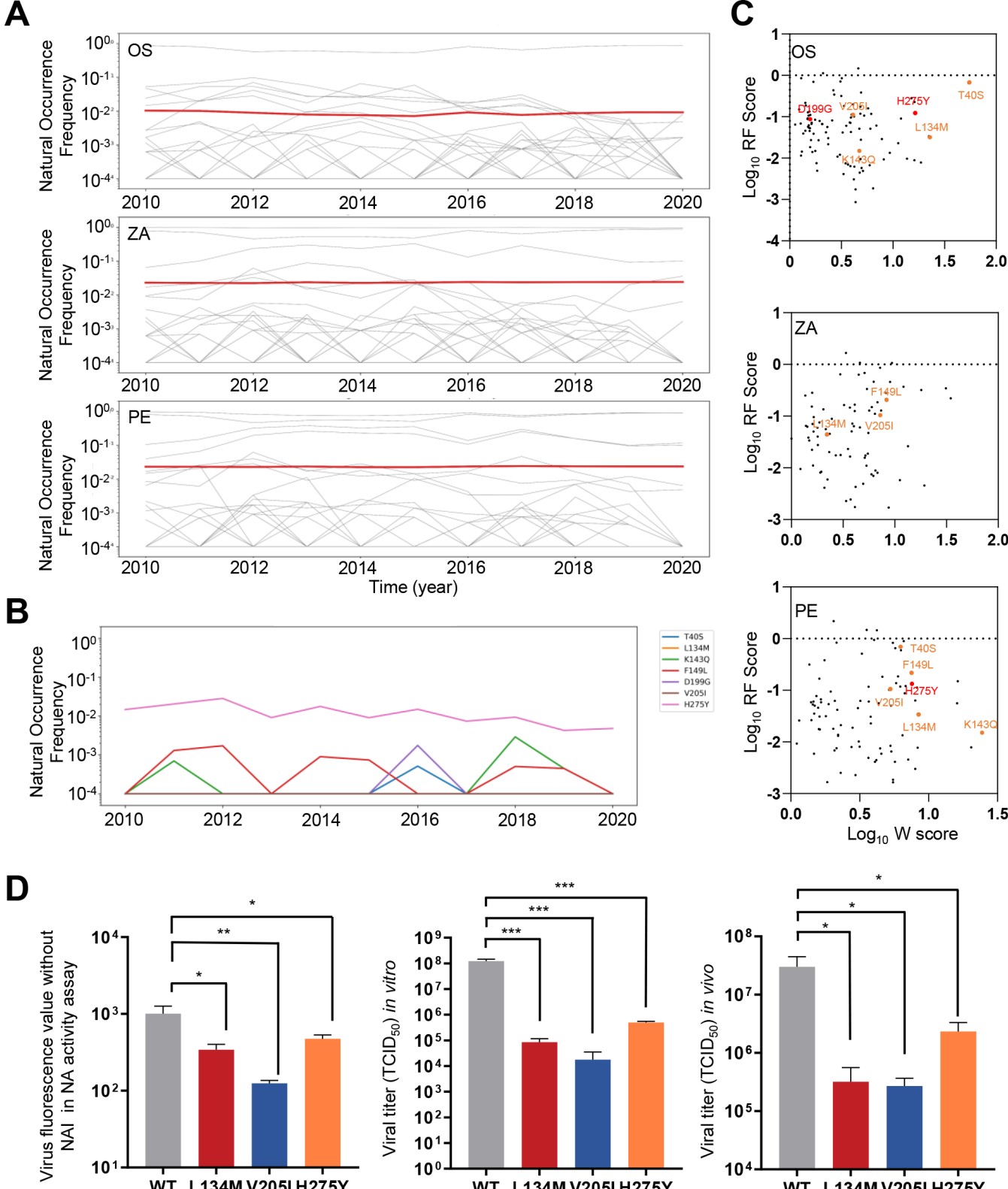

**FIG 5** The replication fitness cost of NAI-resistant mutations may limit their dominance among circulating strains. (A) The natural occurrence dynamics of all identified drug-resistant mutants from 2010 to 2020 (99, 82, and 86 mutations for each drug, respectively). The horizontal red line represents the mean occurrence of these mutants. (B) The time trajectory of the major drug-resistant mutations previously reported or validated in the present study (T40S, L134M, K143Q, F149L, V205I, D199G, and H275Y). (C) The correlation between the W and RF scores of NAI-resistant mutations is shown. W scores were computed under (Continued on next page)

**FIG 5 (Continued)**

OS, ZA, and PE selection conditions. The previously reported drug-resistant mutation H275Y is denoted in red. The newly validated drug-resistant mutations identified in our study are highlighted in orange. (D) Comparison of the viral replication capacity of the four major NAI-resistant mutants in the NA activity assay (left, $N = 3$), the viral growth assay in cell culture (middle, $N = 3$), and the viral growth assay *in vivo* (right, $N = 3$). Error bars denote SD. $*P < 0.05$, $**P < 0.01$, $***P < 0.001$ (two-tailed *t*-test).

limited permissive cells, multiple types of innate immune cells, and an antiviral mucosal environment. To overcome this, we controlled the complexity of each library, used a high dose of infection, and administered NAIs at 6 h post-infection. While utilizing viruses reconstituted from 293T cells would have circumvented the selection bias in MDCK cells, we utilized the amplified virus from MDCK as it provided sufficient viral titer for *in vivo* studies and a consistent linkage between genotype and phenotype. Moreover, due to the permissiveness of MDCK cells to IAV infection, the one-round amplification did not hinder our ability to analyze the differential fitness effects of mutations between *in vivo* and *in vitro* conditions, as depicted in Fig. 2E and F. Furthermore, the high correlation observed among RF scores in biological replicates (Fig. 1B) demonstrates our system's reproducibility and reliability.

The dissemination of drug-resistant mutations is primarily governed by their effects on drug resistance and viral fitness. The high-throughput genetic approach offers a systematic solution to simultaneously assess relative fitness and drug resistance, which is informative for predicting potential circulating strains in the future (52, 53). It alleviates the reliance on the emergence of beneficial mutations in long-term evolution experiments starting from WT viruses, which can be stochastic and time-consuming. By applying this platform to an *in vivo* drug selection system, we successfully captured previously reported mutations and novel substitutions in influenza NA that confer NAI resistance.

Most NAI-resistant mutations have been reported to occur either in the catalytic domain or in framework residues close to the NA enzyme active site (53–55). Consistently, the newly found drug-resistant mutation in our screen, L134M, is located at one interior hydrophobic core of NA. The side chain of the mutation L134M may reach out to the binding pocket and disrupt the drug deposition. Furthermore, our study revealed that mutations occurring outside the catalytic site of NA, such as V205I, can confer reduced sensitivity to NAIs, unveiling a different mechanism for NA drug resistance. This particular mutation resides at the interface between two NA molecules, where it has the potential to weaken the interaction and increase the local dynamics of the NA structure. Although these mutations exhibited low relative fitness in the tested A/WSN/33 background, potentially limiting their evolution, it is crucial to consider the possibility of compensatory mutations that could arise and restore the growth of these drug-resistant mutants. Therefore, these variants' potential public health threats should not be overlooked.

Though we successfully profiled the mutational effect of NA and drug-resistant mutations on three NAIs, our method still had some limitations. Firstly, using random mutagenesis to introduce mutations for the NA segment, only the frequency of single nucleotide mutations was high enough in our libraries for quantitative measurement, although mutations of multiple nucleotides exist. Consequently, we could not explore the effects of multiple nucleotide mutations. Secondly, the reconstituted viral library underwent one round of amplification in MDCK cells prior to *in vivo* screening. This process may lead to a reduced relative frequency of certain deleterious mutations. Consequently, there is a potential for inaccuracies in calculating these deleterious mutations' relative drug resistance score (W), which could result in false positives in detecting drug resistance mutations. Finally, we constructed the NA library using the A/WSN/33 (H1N1) virus. Different genetic backgrounds may exhibit varying profiles of drug-resistant mutations, particularly when compensatory mutations are involved (56–59). While we anticipate that most of our conclusions will hold true for other strains, conducting comparative studies using different genetic backgrounds is essential.

## MATERIALS AND METHODS

### Compounds

The NAIs, including OS, ZA, and PE, were purchased from Meilunbio (China).

### Viruses, cells, and mice

Influenza A/WSN/33 WT and the mutant viral libraries were generated using the eight-plasmid reverse genetics system (60). HEK-293T (293T) cells were cultured in DMEM (Dulbecco's Modified Eagle Medium, Gibco) with 10% FBS (Fetal bovine serum, Biological Industries). MDCK cells were cultured in DMEM with 10% FBS. A549 cells were cultured in RPMI 1640 (Gibco) with 10% FBS. Female BALB/c mice (SLACOM) at 6–8 weeks were sacrificed for the screening. All mouse experiments were performed following the guidelines of the animal protocols approved by Zhejiang University, China.

### Construction of an influenza NA mutant plasmid library

Influenza A/WSN/33 NA mutant plasmid libraries were generated as previously described (49). In brief, the influenza NA gene (29–1,228 bp) was separated into five small fragments of 240 bp each. Random mutagenesis was performed for each fragment separately with the error-prone polymerase Mutazyme II (Stratagene). The following primers were used: CAGGTCTCATAAATGAATCCAAACCAGAAAAT (flu6-M1F-BsaI), CAGGTCTCATGGAACCAATTCTTATGCCATTG (flu6-M1B-BsaI), CAGGTCTCAGGG-TGGGCTATACACAGCAAAGA (flu6-M2F-BsaI), CAGGTCTCAGCCAGCCCATTCCATCATGACAT (flu6-M2B-BsaI), CAGGTCTCATCGGTTGCTTGGTCAGCAAGTGC (flu6-M3F-BsaI), CAGGTC-TCACATTCAACTCTATCGATTTAGTA (flu6-M3B-BsaI), CAGGTCTCATTCAAGATCGAGAAG-GGGAAGGT (flu6-M4F-BsaI), CAGGTCTCAATGAAAATCCCTTTACTCCGTTT (flu6-M4B-BsaI), CAGGTCTCAGGCCCAGTGTCTGCTGATGGAGC (flu6-M5F-BsaI), and CAGGTCTCAGTAGCC-CCCTGATTAATTCAACC (flu6-M5B-BsaI).

The corresponding vectors for each of the five mutant libraries were generated by PCR using the following primers: CACGTCTCATTTAAACTCCTGCTTTCGCTCCC (flu6-V1F-BsmBI), CACGTCTCTTCCAAAGGAGATGTTTTTGTCAT (flu6-V1B-BsmBI), CACGTCTCCACC-CACGGATGGGACAAAGAGAT (flu6-V2F-BsmBI),

CACGTCTCCTGGCTAACAATCGGAATTTCTGG (flu6- V2B-BsmBI),

CACGTCTCACCGATTCAAACCTTGAATTGTAC (flu6- V3F-BsmBI), CACGTCTCGAATGCA-CCTAATTCTCACTACGA (flu6- V3B-BsmBI), CACGTCTCTTGAAAATTTTGTACGAGGCCAGC (flu6- V4F-BsmBI), CACGTCTCTTCATATAAGTATGGCAATGGTGT (flu6- V4B-BsmBI), CACGTCTCGGGCCACAGCTGCCTGTTCCATCT (flu6- V5F-BsmBI), and CACGTCTCGCTACCT-GAGGAGGACGCAATCTG (flu6- V5B-BsmBI).

The PCR product was digested by DpnI (New England Biolabs) to remove the input plasmid.

The amplified segment was gel purified for each small library, BsaI digested, ligated to the BsmBI digested vector, and transformed into MegaX DH10B T1R cells (Life Technologies). As each small library was expected to have ~1,000 single mutations, ~50,000 bacterial colonies were collected to ensure ~15× coverage for each mutant (considering ~33% of the library is single-nucleotide mutations). The plasmid libraries were extracted using PureLink HiPure Plasmid Filter Midiprep Kit (Invitrogen).

### Library reconstitution and amplification

To generate the mutant viral library, ~30 million 293T cells were transfected with each plasmid library (1 out of 5 small libraries) and the other 7 plasmids encoding WT viral proteins (72 µg of DNA in total). Transfections were performed using Hieff TransTM Liposomal Transfection Reagent (Yeasen). The medium was replaced at 12 h post-transfection. Viruses were collected 60 h post-transfection. Viral titers ($TCID_{50}$) were measured using MDCK cells. To further amplify the mutant viral libraries, ~10 million MDCK cells were infected with the individual mutant virus libraries at an MOI of 0.1 (library

coverage ~300-fold). Cells were washed with PBS three times at 2 h post-infection. Viruses were collected 48 h post-infection from the supernatants.

## Intranasal infection of mice with mutant libraries and drug selection

We determined the dose-response curve of each drug to ensure that the selection pressure was consistent among the three drugs. We selected a relatively high drug concentration (IC90 for each drug: OS 1 mg/kg/d, ZA 0.1 mg/kg/d, and PE 1 mg/kg/d). Groups of 6- to 8-week-old female BALB/c mice (SLACOM) ($N$ = 3/group as biological replicates) were lightly anesthetized with inhaled isoflurane and then inoculated intranasally with $1.0 \times 10^5$ $TCID_{50}$ of the NA mutant viral libraries in 50 µL (library coverage ~30-fold). At 6 h post-infection, NAIs were administered (OS orally, ZA intranasally, and PE intramuscularly) to each mouse group. The control group of mice was left untreated in parallel. The animals were monitored daily for clinical signs of disease, body weight changes, and survival and were administered three NAIs daily for 3 days post-infection. The dosage was the same as that on day 1. Mice that lost over 30% of their initial body weight were euthanized. Then, the lung of each mouse was harvested on day 3 post-infection.

## A549 cells infected with mutant libraries for *in vitro* screening

To screen viral libraries without NAI treatment, ~10 million A549 cells were infected with individual mutant virus libraries at an MOI of 0.01. Cells were washed with PBS three times at 2 h post-infection. Viruses were collected 48 h post-infection from the supernatants, cleared of cell debris, and stored at −80°C in aliquots.

## RNA extraction, reverse transcription, and real-time PCR

Each mouse lung sample was mixed with 1 mL of TRIzol reagent (Cwbiotech) and homogenized at 4°C. Viral RNA was extracted using TRIzol reagent and reverse transcribed using HiScript II Q RT SuperMix for qPCR (Vazyme). Quantitative real-time PCR was performed using 2× T5 Fast qPCR Mix (SYBR Green I) (Tsingke). For viral copy number, a series of standard samples ranging from $1 \times 10^3$ to $1 \times 10^8$ copies per µL was used for quantification. The primer sequences used to quantify RNAs are GAC GAT GCA ACG GCT GGT CTG (forward) and ACC ATT GTT CCA ACT CCT TT (reverse).

## Mutant library sequencing and data analysis

For each mouse sample, at least $1 \times 10^6$ genome copies were used to amplify the mutated segment. The primers used for amplification are as follows: GAGTTT-CTGGAGATCCAAACCAGAAAAT (flu6-S1F-BpmI), TCCTTTCTGGAGAATTCTTATGCCATTG (flu6-S1B-BpmI), TCCGTGCTGGAGCTATACACAGCAAAGA (flu6-S2F-BpmI), TGTTAGCTG-GAGCATTCCATCATGACAT (flu6- S2B-BpmI), TTGAAT CTGGAG CTTGGTCAGCAAGTGC (flu6- S3F- BpmI), AGGTGC CTGGAG CTCTATCGATTTAGTA (flu6- S3B-BpmI), AAATTT CTGGAG TCGAGAAGGGGAAGGT (flu6- S4F- BpmI), CTTATA CTGGAG TCCCTTTACTCCGTTT (flu6- S4B-BpmI), GCTGTG CTGGAG TGTCTGCTGATGGAGC (flu6- S5F-BpmI), and CTCAGG CTGGAG CCTGATTAATTCAACC (flu6- S5B-BpmI).

The amplified regions were BpmI digested and ligated with the sequencing adaptor, which had a three-nucleotide multiplexing ID to distinguish between samples.

Deep sequencing was performed with Illumina sequencing NovaSeq PE250. The three-nucleotide population IDs were used to de-multiplex raw sequencing reads. The same table of the multiplexing index for each sample is provided in Table S3. We filtered out the reads with a quality score of less than 30 and a mismatch between forward and reverse reads. After filtering, the average read depth for each library is ~400,000, and the minimal read depth is ~200,000, ensuring enough coverage for each mutant.

Sequencing errors were corrected by filtering out unmatched forward and reverse reads. Mutations were identified by comparing sequencing reads with the WT sequence.

The RF score was calculated for each point mutation, and only mutations with frequencies above 0.01% in the plasmid library were reported.

$$RF\ Score_{mutant\ i} = \frac{Relative\ Frequency\ of\ Mutant\ i_{infection}}{Relative\ Frequency\ of\ Mutant\ i_{plasmid}}$$

where

$$Relative\ Frequency\ of\ Mutant\ i = \frac{Reads\ of\ Mutant\ i}{Reads\ of\ WT}$$

For a particular drug, each mutation's drug resistance score (W) was computed as the proportion of the RF score with drug selection to that without drug selection.

$$W\ Score_{mutant\ i} = \frac{RF\ Score\ of\ Mutant\ i_{with\ drug}}{RF\ Score\ of\ Mutant\ i_{without\ drug}}$$

## Classification of functional residues, essential residues, and drug-resistant mutations

We define a residue as functional for viral replication if more than 50% of its mutations have RF <0.1 and ΔΔG <0. A residue is essential if the frequency of its mutations with RF <0.1 is >80%. Under each drug selection condition, drug-resistant mutations include those with a significantly higher W than synonymous mutations (two-tailed $t$-test, $P <$ 0.001), or multiple mutations in one residue with min W >1.5 for all three replicates.

## Predictions of protein stability

The protein stability of a single mutation on NA (ΔΔG) was predicted by PyRosetta (32). The PDB file 3B7E was cleaned and trimmed to single-chain atoms. Then, all side chains were repacked and minimized using the score function ΔΔG monomer. Then, we introduced all possible substitutions within the structures and repacked all atoms within 8 Å using the Linmin mover. The new structure was scored again, and the difference between the new score and the score of the initial variant was used as ΔΔG. The procedure was repeated 10 times, and the mean results were employed in the correlation analysis.

## Diversity of IAV sequences identified in IRD

Amino acid sequences of the influenza NA protein were downloaded from the IRD. IRD is an open online resource funded by the US NIH/NIAID that facilitates the retrieval, analysis, and visualization of influenza virus data. Shannon entropy was used to quantify the sequence diversity at each amino acid site.

## Construction of viruses with a specific mutation

Individual mutant viral plasmids were generated by a PCR-based site-directed mutagenesis strategy. To generate a single mutant virus, ~2 million 293T cells were transfected with 4 µg DNA. Transfections were performed using Hieff TransTM Liposomal Transfection Reagent (Yeasen). The medium was replaced at 12 h post-transfection. Viruses were collected 60 h post-transfection. Viral titers (TCID$_{50}$) were measured with MDCK cells. To amplify the mutant viruses further, ~10 million MDCK cells were infected with individual mutant viruses at an MOI of 0.1. Cells were washed with PBS three times at 2 h post-infection. The viruses were collected 48 h post-infection from the supernatants. To confirm the genotype of the mutant viruses, each mutant viral RNA was extracted and reverse-transcribed to cDNA. Each mutant's sequence was PCR amplified and then subjected to Sanger DNA sequencing. To measure the growth kinetics of mutant viruses

with and without NAI selection, ~0.2 million A549 cells were pre-treated with NAIs for 2 h, respectively, and then infected with each mutant virus at an MOI of 0.01 with/without NAIs, and the supernatants were collected at 48 h post-infection.

## Quantification of the relative replication capacity of mutants *in vivo*

We evaluated the relative replication capacity of mutants by mixing each mutant with WT viruses at an equal ratio and performing intranasal infection on mice. Groups of 6- to 8-week-old female BALB/c mice (SLACOM) ($N = 3$/group as biological replicates) were lightly anesthetized with inhaled isoflurane and then inoculated intranasally with $1.0 \times 10^5$ TCID$_{50}$ of the mixed viruses in 50 µL. Lung tissues were harvested at day 3 post-infection, viral RNA extracted, reverse transcribed, amplified with the input viral mix, and subjected to Sanger sequencing. Relative peak heights of mutant and WT nucleotides at each position were quantified.

## Examination of IFN sensitivity

For confirmation of IFN sensitivity, A549 cells were pre-treated with 1,000 U/mL exogenous IFN (IFNa2, 1,000 U/mL) (PBL Assay Science) for 20 h and then infected with individual mutants or WT virus at an MOI of 0.01. Cells were washed with PBS three times at 2 h post-infection, and 1,000 U/mL IFNa2 was added back into the culture medium. The viral copy number was examined 24 h post-infection. IFN sensitivity was calculated as the decreased viral copy number upon IFN treatment.

## Neuraminidase activity assay

We measured the NA activity in recombinant viruses using a fluorometric assay with the fluorogenic substrate MUNANA (Sigma-Aldrich, Inc., St. Louis, MO, USA) (61). Briefly, each recombinant virus was normalized to the same NA activity without the drug and incubated with different concentrations of NAIs for 45 min. The incubation proceeded for 1 h with the fluorogenic substrates MUNANA and NAIs. Afterward, the stop solution was added to stop the reaction, and the fluorometric value was quantified with Thermo Scientific Varioskan Flash. The concentration of NAIs (OS, ZA, and PE) ranging from 0.0003 to $1 \times 10^6$ µM for 50% inhibition of the NA activity (IC50) was calculated from the fitted dose-response curve using GraphPad Prism 8.3.0 software (La Jolla, CA, USA).

## Recombinant protein expression and purification

NA WT and mutant recombinant proteins were produced by the Bac-to-Bac baculovirus expression system. Protein coding sequences were cloned into pFASTBAC-1 vectors with an N-terminal GP67 signal peptide and a C-terminal His-tag. Baculoviruses were produced by sf9 insect cells. Proteins were expressed by High5 insect cells and secreted into the media. The culture medium was collected when the cell viability was ~10%. The secreted proteins were purified by Ni-column and Superdex 200 (GE Healthcare). Fractions containing target proteins were pooled and concentrated.

## Isothermal titration calorimetry

To test the effects of the mutations on NA proteins, we measured the binding affinities between the drug and purified NA proteins by ITC. The tests were performed at 25°C (MicroCal iTC200, Malvern Panalytical). The buffer contained 20 mM MES (pH = 6.5) and 200 mM NaCl. The protein concentration in the syringe was around 0.006 mM, while in the reaction cell, it was around 0.125 mM. All titration data were calculated and analyzed by MicroCal ITC-ORIGIN Analysis Software (Malvern Panalytical).

## ACKNOWLEDGMENTS

The study was supported by grants from the National Natural Science Foundation of China (82272300, 82102893, and 82202488) and the Science Technology Department of Zhejiang Province (2019C03018).

## AUTHOR AFFILIATIONS

[1]Cancer Institute (Key Laboratory of Cancer Prevention and Intervention, China National Ministry of Education), The Second Affiliated Hospital, Zhejiang University School of Medicine, Hangzhou, Zhejiang, China

[2]Affiliated Hangzhou First People's Hospital, Zhejiang University School of Medicine, Hangzhou, China

[3]Molecular Biology Institute, University of California, Los Angeles, California, USA

[4]Department of Surgery, Women's Hospital, School of Medicine, Zhejiang University, Hangzhou, China

[5]Department of Molecular and Medical Pharmacology, University of California, Los Angeles, California, USA

[6]School of Biomedical Sciences, LKS Faculty of Medicine, The Hong Kong University, Hong Kong, China

[7]Department of Ultrasound in Medicine, The Second Affiliated Hospital of Zhejiang University School of Medicine, Hangzhou, China

[8]Department of Medical Oncology, The Second Affiliated Hospital of Zhejiang University, School of Medicine, Hangzhou, China

[9]Pharmaceutical Informatics Institute, College of Pharmaceutical Sciences, Zhejiang University, Hangzhou, China

[10]Department of Colorectal Surgery and Oncology, Key Laboratory of Cancer Prevention and Intervention, Ministry of Education, The Second Affiliated Hospital, Zhejiang University School of Medicine, Hangzhou, Zhejiang, China

[11]Center for Infectious Disease Research, Westlake Laboratory of Life Sciences and Biomedicine, Hangzhou, Zhejiang, China

## AUTHOR ORCIDs

Ren Sun http://orcid.org/0000-0002-8628-7019
Yushen Du http://orcid.org/0000-0002-8015-4207

## DATA AVAILABILITY

All research materials are available upon request. Raw sequencing data have been submitted to the NIH Sequence Read Archive (SRA) under accession number: BioProject PRJNA801090. All scripts for data analysis have been deposited at https://github.com/Tian-hao/flu_NA.

## ADDITIONAL FILES

The following material is available online.

### Supplemental Material

**Fig. S1 (mSystems00670-23-s0001.tif).** Construction of NA single nucleotide mutation libraries.
**Fig. S2 (mSystems00670-23-s0002.tif).** Comparison of fitness profiling *in vivo* and *in vitro*.
**Fig. S3 (mSystems00670-23-s0003.tif).** Profiling of drug-resistant mutations in NA *in vivo*.
**Fig. S4 (mSystems00670-23-s0004.tif).** Validation of NAI-resistant mutants.
**Supplemental legends (mSystems00670-23-s0005.docx).** Legends to Fig. S1-S4 and Tables S1 to S3.

**Table S1 (mSystems00670-23-s0006.xlsx).** The *in vivo* RF score and W score of each mutation of NA.

**Table S2 (mSystems00670-23-s0007.xlsx).** Mutations showing resistance against three NAIs.

**Table S3 (mSystems00670-23-s0008.xlsx).** Multiplexing index for each sample.

## Open Peer Review

**PEER REVIEW HISTORY (review-history.pdf).** An accounting of the reviewer's comments and feedback.

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
