## [Reviewer comments · mSystems]

Deep mutational scanning of Influenza A virus neuraminidase facilitates the identification of drug resistance mutations in vivo

sihan wang, Tianhao Zhang, Menglong Hu, Kejun Tang, Li Sheng, Mengying Hong, Dongdong Chen, Liubo Chen, Yuan Shi, Jun Feng, Jing Qian, Lifeng Sun, Kefeng Ding, Ren Sun, and Du Yushen

Corresponding Author(s): Du Yushen, Zhejiang University

Review Timeline:

Submission Date:

July 3, 2023

Accepted:

August 9, 2023

Editor: Jack Gilbert

Reviewer(s): The reviewers have opted to remain anonymous.

Transaction Report:

DOI: <https://doi.org/10.1128/msystems.00670-23>

August 9, 2023

Dr. Du Yushen
Zhejiang University
Hangzhou
China

Re: mSystems00670-23 (Deep mutational scanning of Influenza A virus neuraminidase facilitates the identification of drug resistance mutations in vivo)

Dear Dr. Du Yushen:

Your manuscript has been accepted, and I am forwarding it to the ASM Journals Department for publication. For your reference, ASM Journals' address is given below. Before it can be scheduled for publication, your manuscript will be checked by the mSystems production staff to make sure that all elements meet the technical requirements for publication. They will contact you if anything needs to be revised before copyediting and production can begin. Otherwise, you will be notified when your proofs are ready to be viewed.

If you would like to submit a potential Featured Image, please email a file and a short legend to msystems@asmusa.org. Please note that we can only consider images that (i) the authors created or own and (ii) have not been previously published. By submitting, you agree that the image can be used under the same terms as the published article. File requirements: square dimensions (4" x 4"), 300 dpi resolution, RGB colorspace, TIF file format.

We recognize that the video files can become quite large, and so to avoid quality loss ASM suggests sending the video file via <https://www.wetransfer.com/>. When you have a final version of the video and the still ready to share, please send it to mSystems staff at msystems@asmusa.org.

Sincerely,

Jack Gilbert
Editor, mSystems

Journals Department
E-mail: mSystems@asmusa.org